# How Does Agricultural Trade Liberalization Have Environmental Impacts? Evidence from a Literature Review

Panxian Wang [1], Zimeng Ren [2] and Guanghua Qiao [1,*]

1   College of Economics and Management, Inner Mongolia Agricultural University, Hohhot 010010, China; pxwang@emails.imau.edu.cn
2   College of Economics and Management, Northeast Agricultural University, Harbin 150030, China
*   Correspondence: qiao_imau@126.com

**Abstract:** The liberalization of world trade has led to a significant increase in agricultural trade, which has brought to light various environmental externalities, including climate change, deforestation, and water pollution. While economic studies tend to overlook the environmental effects of agricultural trade liberalization, recent research has shown a growing interest in related aspects. As such, it is crucial to conduct a comprehensive analysis of the environmental impacts of agricultural trade liberalization. This study aims to address this issue by conducting a systematic review of the relevant literature from the past two decades. Research has revealed that agricultural trade liberalization has both positive and negative impacts on the environment. The various mechanisms through which these effects are observed include scale, structural, transport, and technology effects. Most studies have concluded that agricultural trade liberalization has a significantly negative impact on the environment. To address this issue, four potential solutions have been proposed, including factor allocation, policy adjustment, technological innovation, and improvements to compensation mechanisms. Future research should aim to develop a comprehensive model that can effectively examine the environmental impacts of agricultural trade policy distortions and the criteria used to select environmental measures. By doing so, we can gain a deeper understanding of the complex relationship between agricultural trade policies and their environmental consequences.

**Keywords:** agricultural trade liberalization; environmental externalities; climate change; mechanisms of action; literature review

## 1. Introduction

In recent years, global food systems have faced significant challenges due to the growth of the global population, the COVID-19 pandemic, frequent natural disasters, and the intensification of regional conflicts. As a result, food security has been greatly impacted. Agriculture, which serves as the backbone of global food security, utilizes 70% of the world's freshwater, 11% of its land surface, and contributes to 22% of global anthropogenic greenhouse gas emissions [1]. Despite its crucial role, agriculture is confronted with significant challenges, including the diversification of human needs and urbanization. Food systems and people are interconnected like a chain through trade. The process of moving food from areas with a surplus to areas with a deficit not only helps to balance the global food supply and demand, but also provides better access to foreign markets and increases the exportation of agricultural products [2]. As a result, employment opportunities are created and expanded, and farmers' incomes are raised. Therefore, trade is crucial in satisfying the needs of worldwide food consumers who seek a varied and healthy diet.

Since the beginning of the 21st century, the agricultural trade sector has witnessed significant growth largely attributed to the implementation of trade liberalization measures, such as the reduction of food tariffs, the Uruguay Round negotiations, and the signing of multiple agreements. It is noteworthy that the market share of emerging economies,

including China and Brazil, has significantly increased, thereby playing a pivotal role in global agricultural and food markets [2]. In 2021, global agricultural export trade exceeded $200 billion for the first time, reaching a staggering $2,162,376 million (these figures are calculated from the UN comtrade database, https://comtradeplus.un.org/, accessed on 3 December 2022). Interestingly, the proportion of exports from low- and middle-income countries has risen from approximately 30% in 1995 to 40%, whereas high-income countries have sustained their 60% share since that time [2].

Trade and the environment are two crucial issues that are closely intertwined. However, the connection between trade liberalization and environmental quality is highly intricate. Even today, there is no academic consensus on the impact of trade liberalization on the environment, and the debate continues. Nevertheless, the existing literature acknowledges that trade liberalization can have both positive and negative effects on the environment [3,4]. International trade has been found to have a significant impact on the environment, leading to the creation of environmental externalities. The liberalization of trade has further exacerbated this issue, particularly in the agricultural sector. As a result, environmental issues, such as climate change [5], the depletion and pollution of freshwater resources [6,7], the eutrophication of river bodies [8], tropical deforestation [9], and biodiversity loss have become more prominent [10,11]. Studies have shown that these externalities are directly linked to the expansion of agricultural trade. Trade can also promote production in the most efficient areas, ultimately reducing reliance on natural resources for agriculture.

The impact of agricultural trade liberalization on the environment is multifaceted and dependent on various factors. These include the economic stage of countries and their natural conditions, agricultural status, and comparative advantages.

The environmental impacts of agricultural trade liberalization are complex and depend on local conditions. Weak regulation can lead to environmental externalities, such as unsustainable groundwater abstraction and land degradation, which can have negative effects on local or regional ecosystems. However, the most challenging and difficult-to-manage impacts are global externalities, including ecosystem destruction and greenhouse gas emissions. Agriculture plays a significant role in contributing to global greenhouse gas emissions and other environmental impacts. Drabo [12] highlights that the production of major commodity exports in agriculture is responsible for further increasing these emissions. Statistics show that direct greenhouse gas (GHG) emissions from agriculture, forestry, and other land uses make up 22% of global emissions. It is important to note that when agricultural production or deforestation takes place in a particular region, the resulting effects of climate change also occur far from where GHGs are released into the atmosphere [13].

Agricultural trade liberalization entails the reduction of policies that distort trade and the encouragement of productive investments and technological advancements. These are the key drivers of sustainable agricultural growth, leading to a significant increase in food and agricultural trade across all regions in a hypothetical scenario of global frictionless trade. The growth in revenue from trade has the potential to increase the demand for policies that promote the production of environmental goods. Additionally, it can incentivize regulators to manage their resources more carefully, leading to more effective environmental management. Free trade has the potential to enhance production efficiency, minimize energy consumption, and facilitate access to novel technologies.

Although economic studies have traditionally neglected the environmental consequences of agricultural trade liberalization [14], research on this topic has significantly expanded since the start of the 21st century. As global, national, and regional concerns for the environment continue to grow, it is crucial to conduct an in-depth analysis of the environmental effects of agricultural trade liberalization [15]. This highlights the continued importance of studying this topic. This paper conducts a systematic review based on the relevant literature of the last two decades to answer several questions regarding agricultural trade liberalization and its impact on the environment. Firstly, it explores whether agri-

cultural trade liberalization affects the environment and under what conditions. Secondly, it examines the mechanisms behind this impact. Thirdly, it identifies the types of impacts that can result from agricultural trade liberalization. Fourthly, it suggests ways to address these impacts. Finally, the paper looks to the future by exploring the frontiers of research on agricultural trade liberalization and its environmental impacts.

This paper not only examines the types of environmental impacts of agricultural trade liberalization, but also its impact mechanisms, providing a more comprehensive understanding of the relationship between various environmental effects. Additionally, solutions have been proposed from different perspectives to address the negative impacts of agricultural trade liberalization on the environment, making this study innovative and valuable.

Therefore, this paper is structured as follows: Section 2 introduces the methodology used in this study. Section 3 explains the mechanism by which agricultural trade liberalization effects environmental impacts. Section 4 examines the various environmental impacts of agricultural trade liberalization. Finally, we conclude with a summary of our findings and suggestions for future research.

## 2. Materials and Methods

### 2.1. Journal Sources

To obtain literature that is highly relevant to the environmental impact of agricultural trade liberalization, we employed several methods. Firstly, we conducted a search on the "Web of Science" platform for relevant articles. We only included studies written in English, and we began our literature search with the specific phrase "agricultural trade liberalization and the environment," which had to appear in the title, abstract, or keywords. We limited our search to articles published between 2003 and 2022, covering the last 20 years. Additionally, we focused on identifying literature that was particularly pertinent to our study topic, drawing from the higher-quality search results and their cited sources.

To ensure a thorough and pertinent literature review, the authors employed a meticulous retrospective search strategy. To begin, the papers should be filtered based on their titles. Next, the abstracts should be read to eliminate any that do not meet the necessary criteria and to choose ones that align with the topic of this study. Afterwards, the remaining papers should be closely examined, and relevant references should be located by reviewing their citations. Ultimately, this paper will focus on the combination of both the selected papers and their related references.

### 2.2. Screening Process

We conducted a search on the SCI platform for articles related to "agricultural trade liberalization and environment" and "agriculture, trade, liberalization, environment," limiting the date range to 2003–2022. This search yielded 132 entries, which we carefully reviewed and analyzed for relevance to our subject. After reviewing the titles of relevant papers, 71 items were excluded, leaving 61 items that met the criteria for inclusion in this study. Additionally, 53 high-quality related papers cited by 35 of the selected papers were included, bringing the total number of retained papers to 88. The selection process is visually represented in Figure 1.

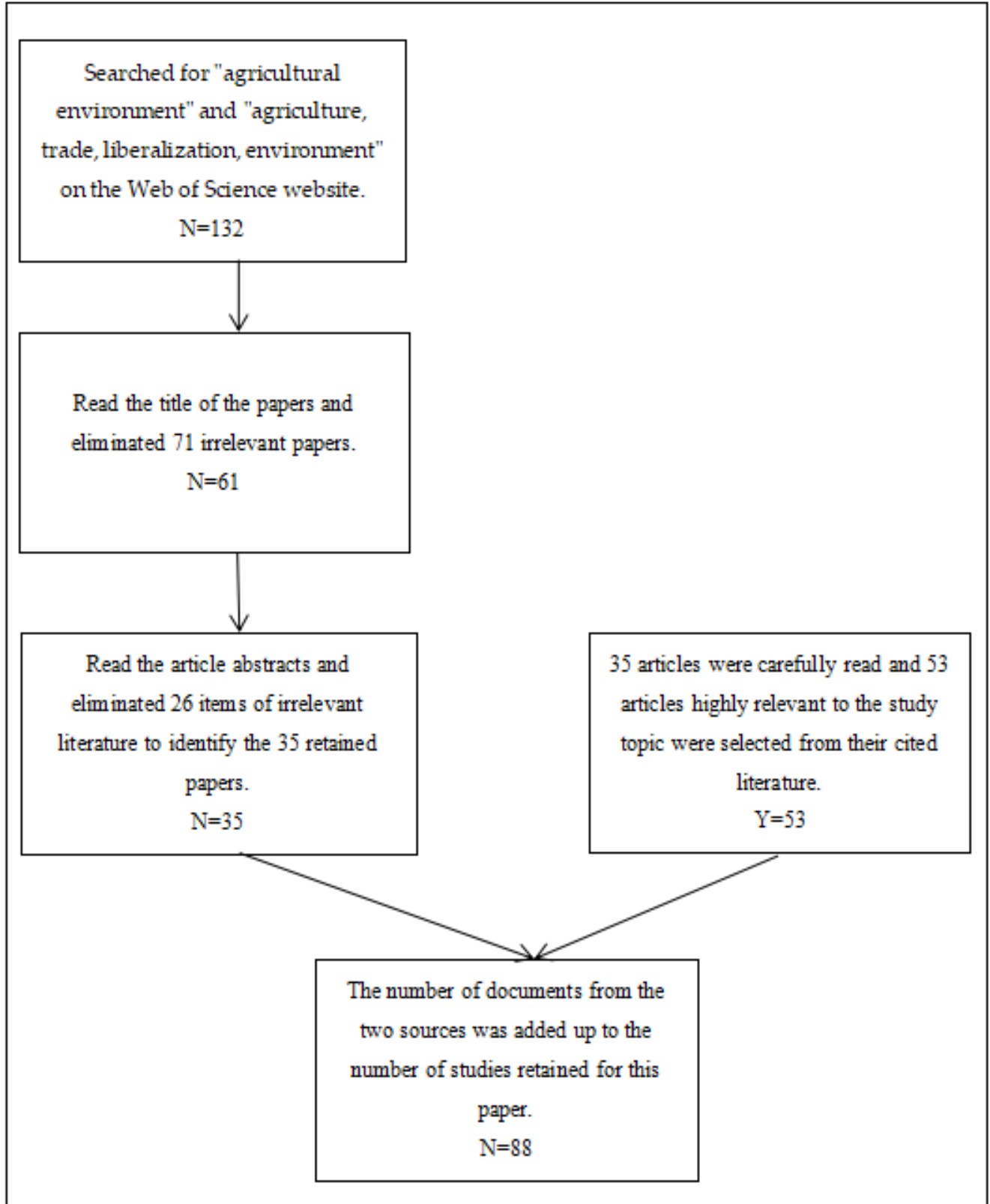

**Figure 1.** Literature screening process.

### 2.3. Literature Classification

In terms of research themes, we can classify the negative environmental impacts of agricultural trade liberalization into several categories. For instance, the production of agricultural products for export may result in unsustainable freshwater extraction, pollution, biodiversity loss, and deforestation. Additionally, the pursuit of high profits with weak regulation can further exacerbate these negative consequences. After conducting a thorough review of the literature, we have identified five primary themes that emerge in the discussion of agricultural trade liberalization and the environment. These themes include the impact of agricultural trade liberalization on climate change, land use, water, deforestation, and environmental standards. Additionally, we found a limited number of papers that address the intersection of agricultural trade liberalization and fisheries.

In terms of country-specific classification, it is crucial to examine the effects of agricultural trade liberalization on the environment in China. As the most populous developing nation, China both imports a diverse range of agricultural products to fulfill the demands of its citizens and exports a significant amount of agricultural goods, which are integral to maintaining global food security. Therefore, studying the impact of agricultural trade liberalization on China's environment is of utmost importance. In recent years, China has made significant efforts towards green and sustainable development, which has important implications for ecological protection. Other countries that have received academic attention include the United States, a long-standing agricultural powerhouse, and Mexico, which is related to NAFTA. Additionally, Asian countries, such as Indonesia and Bangladesh, as well as sub-Saharan African countries, have also been mentioned.

In terms of research methods, empirical modeling is typically considered the most compelling approach and is frequently utilized in scholarly research. As such, it is also the most prevalent method discussed in this paper. Authors often use scenario simulation methods to predict the environmental impacts of agricultural trade liberalization across various scenarios. Some even prefer to utilize case studies to study the environmental impacts of agricultural trade liberalization. Furthermore, this study mentions the use of a systematic literature review. It is essential to build upon previously existing work to advance knowledge. In order to expand our understanding of a subject, it is crucial to identify its boundaries. A comprehensive review of the relevant literature is the key to comprehending the scope and significance of existing research. This paper adopts a literature review approach to investigate the correlation between agricultural trade liberalization and the environment, and to explore potential avenues for future research.

### 3. Mechanisms of the Environmental Impact of Agricultural Trade Liberalization

The impact of agricultural trade liberalization on the environment has been a topic of intense debate among international scholars for many years. However, the relevant literature has shown that agricultural trade liberalization can have both positive and negative effects on the environment. After conducting a thorough review of the literature, it has been determined that the impact of agricultural trade liberalization on the environment can be categorized into three primary components: the scale effect, the structural effect, and the technology effect. This framework is currently the predominant method for analyzing the environmental consequences of trade. However, in this particular study, we are also including the transport effect as an additional factor to consider.

### 3.1. Transportation Effect

In general, liberalizing agricultural trade will undoubtedly increase trade volume, which, in turn, will lead to an increase in agricultural transportation. Unfortunately, this increase in transportation will result in significant amounts of harmful gas and greenhouse gas emissions. As agricultural trade continues to be liberalized, transportation methods are likely to shift from waterways, which are relatively less polluting, to railroads, roads, and airways, which are more polluting. This shift will further exacerbate the negative environmental impact of these transportation modes. On the contrary, if the cost of transporting

goods and their energy consumption can be kept reasonable, and freight costs include certain environmental costs, then liberalizing agricultural trade could help optimize the agricultural transport structure. This, in turn, could slow down or even suppress the emission of harmful gases resulting from the expansion of trade scale.

### 3.2. Scale Effect

The scale effect is observed when agricultural trade is liberalized, leading to an increase in the scale of production as the market expands and economic activities rise. However, this increase in production comes at a cost to the environment, as agriculture consumes significant amounts of land, water, and agrochemicals. If the structure of agricultural production is not optimized to improve resource efficiency, it can result in resource consumption and environmental pollution. As a result, agriculture has a greater impact on a country's natural resources and environment compared to other industries. This negative impact on the environment can be attributed to the scale effects of agricultural practices. Simultaneously, this will increase the income levels of less developed countries and encourage people to prioritize a high-quality environment. This shift in priorities will have a positive impact on the improvement of national environmental regulations and the transformation of industrial structures. Ultimately, this will lead to technological advancements and structural changes. As per capita income levels rise, the consumption structure is expected to shift towards greater animal food consumption. This change will likely place additional pressure on the agricultural environment.

### 3.3. Structural Effect

During the process of agricultural trade liberalization, the expansion of agricultural markets has resulted in the transfer of products from different countries to a broader region. This has led to a restructuring of agricultural products. The term "structural effect" pertains to the varying levels of division of labor in agricultural production and the corresponding production factors required, which can have varying impacts on agricultural output. Specifically, when the expansion of environmental pollution in industries that benefit from agricultural trade is smaller than the reduction in pollution in industries that do not benefit, agricultural trade openness can have a positive effect on the environment. On the contrary, the effect on the environment is negative. A comparative advantage can be attributed to variations in factor endowments and environmental regulations across different countries. In the present situation, the country gains an advantage due to its comparatively lenient environmental laws in comparison to its trade partners. According to the "pollution paradise" theory, opening up trade will result in the relocation of highly polluting industries to countries with lower pollution standards, thereby transforming developing nations into "pollution havens". The production of "clean agricultural products" is primarily concentrated in developed countries, resulting in a skewed regional distribution. As a result, developed countries are continually increasing the production and export of "clean agricultural products" while importing pollution-intensive goods. This trade pattern creates favorable conditions for improving their agricultural environment. As a result, the impact of agricultural trade liberalization on the environment depends largely on the distribution of comparative advantage among countries. This influence on the environmental effect of agricultural trade liberalization is significant to a certain extent. Although the current domestic literature does not strongly support the concept of the "pollution safe harbor," some evidence has been found through the examination of industry data and econometric issues, such as "endogenous" and "spillover" phenomena. This evidence is supported by Millimet and Martínez, who have further explored these industry data and econometric issues [16,17].

### 3.4. Technology Effect

The technology effect refers to the shift in agricultural production technology resulting from the liberalization of agricultural trade, which has a significant impact on the environ-

ment. Sunge and Ngepah argue that agricultural trade can enhance technological efficiency through the transfer of technology [18]. It is anticipated that technological advancements will yield positive environmental impacts, such as the reduction of pollution and ecosystem damage through the implementation of clean technologies. Additionally, the adoption of new technologies can enhance investment and production efficiency, leading to decreased economic losses for the enterprise and reduced harm to the environment. Thus, with the opening of agricultural trade, the promotion and innovation of agricultural technology has played a positive role in reducing the damage to the environment caused by agricultural products. An efficient, sustainable, and environmentally friendly agricultural system is necessary to ensure long-term food security, and technology plays a key role in addressing the challenges of environmental factors [19,20].

The environmental impact of agricultural trade liberalization is influenced by several factors, including transportation mode, agricultural industrial structure, environmental standards, and technical level. Quantifying this impact is a complex task. Over time, agricultural trade liberalization has had a positive impact on the environmental quality of developed countries, but it has also worsened environmental problems in developing countries. In the context of economic development, it has been observed that as a country progresses, its level of pollution tends to increase in the initial stages. However, as the country reaches the later stages of development and its income levels rise, the quality of the environment tends to improve. The Environmental Kuznets (EKC) theory explains this phenomenon by proposing a relationship between environmental pollution and per capita income that takes the form of an "inverted U"-shaped curve. Essentially, the EKC suggests that carbon emissions tend to rise initially with increasing income, but then fall as income levels continue to rise. This theory has significant implications for understanding the relationship between economic growth and environmental sustainability [21].

This paper argues that agricultural trade liberalization impacts environmental quality in four ways: transportation, scale, structure, and technology. Of these, transportation effects are inevitable and, despite considering environmental costs in transportation expenses, some degree of pollution is unavoidable. Our focus should be on minimizing this pollution. Scale effects can have a detrimental impact on environmental quality, whereas technology effects can have a positive impact. In fact, certain studies have indicated that technology effects may even outweigh the negative impacts of scale effects [22]. The extent of these effects, whether positive or negative, is largely determined by the expansion of dominant industries and the contraction of disadvantaged ones. This paper delves into the impact of agricultural trade liberalization on environmental quality, examining it through four dimensions: transportation, scale, structure, and technology. Figure 2 provides a clear depiction of how agricultural trade liberalization in developing countries affects environmental quality, elucidating the underlying mechanism.

There are interactions between these four effects related to agricultural trade liberalization.

1. The transportation effect and scale effect are intertwined. As demand for agricultural products increases, so does the frequency of transportation, exacerbating the negative impact on the environment, including the soil, water, and atmosphere.
2. The liberalization of agricultural trade has a positive impact on the modernization of agriculture and technological innovation. This results in an increased relationship between scale and technology effects, leading to the growth and output of agricultural production. However, the modernization of agriculture can also have negative consequences, such as putting more pressure on the environment through the excessive use of pesticides, fertilizers, and other chemical substances.
3. In the realm of agricultural production, technological innovation and structural changes are closely linked. As a result, it is important to recognize the ways in which these two factors interact. Specifically, certain species or technologies used in agriculture may have negative impacts on the environment. Therefore, it is crucial to consider the potential risks posed by these elements and to take steps to mitigate them.

4. The structural changes in agricultural production and the transnational circulation of agricultural products have led to the liberalization of agricultural trade. However, this change is likely to be inconsistent and may exacerbate the negative effects of transportation. For instance, the demand for transportation can lead to greater environmental problems, such as air pollution and waste accumulation.

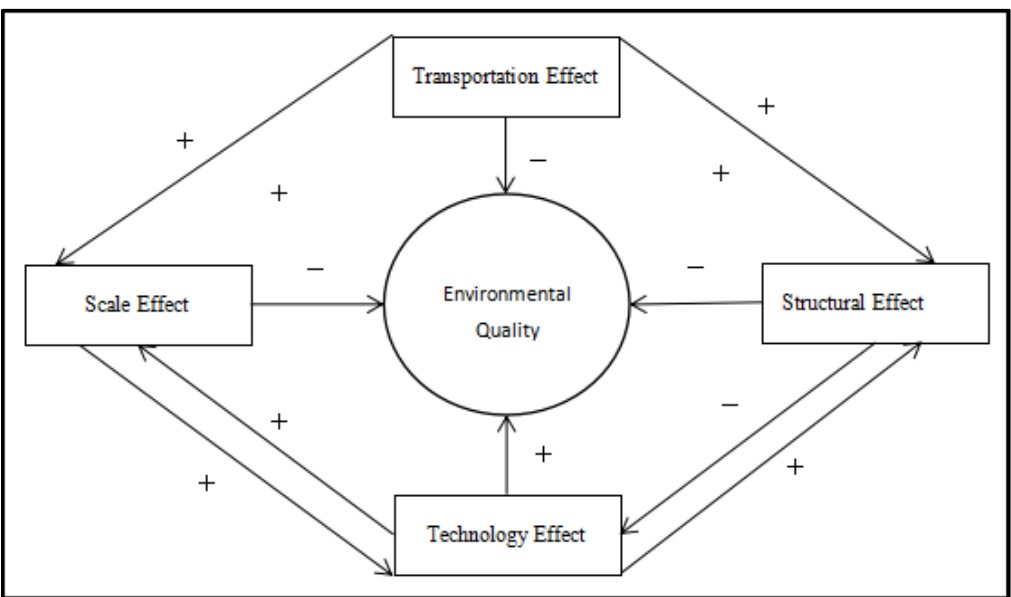

**Figure 2.** Mechanism of environmental impact of agricultural trade liberalization.

## 4. Results

The third section of this paper delves into the impact of agricultural trade liberalization on environmental quality. It becomes clear that quantitatively analyzing this impact poses significant challenges due to the complexity of the factors involved. The environmental quality is influenced not only by the strength of the liberalized sector's production, but also by other factors. Some studies suggest that the presence of sound environmental standards in trading countries can also affect this impact [23]. Currently, there is no empirical method available to provide a thorough analysis of the environmental impact of agricultural trade liberalization. Additionally, discrepancies in the products and sectors studied, the level and type of liberalization, the models and assumptions used, and the environmental indicators chosen can all yield vastly different outcomes. After conducting a thorough literature review, this study suggests that the impact of agricultural trade liberalization on the environment can be categorized into four types. The impact of agricultural trade liberalization on environmental quality is a complex issue with varying outcomes. While some studies suggest that it can lead to an improvement in environmental quality, others argue that it may result in a reduction. Additionally, there is uncertainty surrounding the overall impact of agricultural trade liberalization on environmental quality, and some research suggests that the impact may be insignificant. Overall, it is clear that more research is needed to fully understand the relationship between agricultural trade liberalization and environmental quality.

### 4.1. Agricultural Trade Liberalization Leads to Improved Environmental Quality

Agricultural trade liberalization can have several positive outcomes. It can increase trade and income, facilitate the exchange of goods and services in an environmentally friendly manner, and promote the use of cleaner technologies. Additionally, it can accelerate the diffusion of technology. Two papers from the early 21st century examined the impact of trade liberalization on the environment by using it as a target variable [24,25]. The authors discovered that trade liberalization yields environmental benefits and has a positive impact

on overall environmental quality. This is largely due to the fact that trade liberalization encourages developing countries to adopt clean production processes that were previously only utilized in developed countries.

The impact of free trade on the agricultural environment is not straightforward, and establishing a clear cause-and-effect relationship can be challenging. However, research has shown that agricultural trade liberalization initiated by the WTO can enhance economic efficiency, which in turn leads to better agricultural environmental quality [26]. In their study of forest product trade in 61 countries, Tian et al. discovered that trade liberalization can effectively allocate global timber resources, leading to improved efficiency of utilization and reduced world timber consumption [27]. This, in turn, helps to conserve global forest resources. In their argument, Karunakaran and Sadiq suggest that liberalization and modernization could potentially harm agriculture [28]. However, they propose that organic agriculture is actually beneficial to the environment. They also believe that free trade could support the sustainable development of organic agriculture by providing farmers with long-term crop income and by assisting disadvantaged organic farmers. This idea is supported by their research on the topic [28].

Environmental regulations related to trade can be instrumental in managing the level of environmental damage and promoting sustainable technologies. For the regulations to be effective, it is important to comprehensively integrate environmental and trade agreements at both national and international levels. This integration is crucial in order to improve environmental quality and to maximize the benefits of trade liberalization [29].

Blandford discovered that the Doha Round proposal, aimed at creating a new agreement on agriculture through the World Trade Organization, would not result in significant emission reductions. However, he also found that additional trade liberalization could contribute to a reduction in carbon emissions by decreasing agricultural production [30]. Twerefou et al. hold the view that a higher degree of trade openness is linked to increased emissions due to structural effects, but ultimately leads to a noteworthy decrease in per capita $CO_2$ emissions [31].

Numerous papers have shown that trade liberalization can lead to a decrease in chemical usage. Evan suggests that trade liberalization promotes crop diversity, which in turn can lead to stronger resistance against pest outbreaks and, ultimately, a reduced need for agrochemicals [32]. Rae and Strutt discovered that agricultural trade reform led to a decrease in highly protected agriculture in Western Europe and Northeast Asia, resulting in a reduction in the use of agrochemicals [33]. Zhang and Huang demonstrated that agricultural trade has both scale and structural effects that can lead to increased use of fertilizers and pesticides [34]. However, they also noted that technology effects can actually reduce the need for these inputs. Furthermore, the authors suggest that trade liberalization may have the potential to enhance the overall sustainability of the agricultural sector.

Marta and other experts are concerned that changes in trade patterns could put water resources at risk [35]. However, compared to self-sufficient countries, trade can actually save up to 40–60 cubic meters of water per person per year. Research by Konar shows that Africa's domestic trade is particularly efficient in terms of water resource management [36]. Open trade policies can also lead to a reduction in agricultural water consumption and overall resource use. This is particularly important in Africa, where internal production systems may be lacking. Dang and Konar support this view [37]. On the other hand, Alvaro's research suggests that trade liberalization has little impact on agricultural production and water use [38]. However, it can reduce water consumption in areas with scarce water resources (such as the Middle East and North Africa) while increasing usage in areas with more abundant water resources.

According to Keita et al., fisheries experience a greater reduction in the amount of fish caught at the national level on average as trade openness increases over time [39].

In their analysis of climate change impacts, Baker et al. highlight the significance of free trade in mitigating regional productivity shocks [40]. They achieve this by utilizing global agroforestry models to examine the effects on U.S. agriculture and the rest of the

world. Meanwhile, Cui et al. contend that international trade cannot replace environmental policy [41]. However, they also assert that it should not be considered harmful to environmental outcomes.

### 4.2. Agricultural Trade Liberalization Reduces Environmental Quality

Globally, the environmental challenges resulting from agricultural trade liberalization are widely acknowledged [42]. Given the prevalence of trade liberalization, countries can now rely on environmental policies in lieu of strategic trade policies. However, environmentalists are apprehensive that trade liberalization may cause "ecological dumping" [14]. Weinzettel et al. and Nesme et al. have highlighted the negative impact of trade expansion, stating that the liberalization of agricultural trade will worsen a country's overuse of natural resources [43,44]. On the one hand, agricultural trade liberalization will lead to the expansion of agricultural production and cultivated land, resulting in the direct effect of the scale effect. This, in turn, will worsen the depletion of natural resources and environmental degradation in agriculture. On the other hand, this will worsen the environmental degradation of resources through an indirect effect. By attracting more labor to cultivate marginal agricultural land and by increasing the rate of resource utilization, this will further manifest the structural effect. Furthermore, much like the concept of a "pollution paradise," agricultural liberalization can result in the relocation of agricultural production from developed nations with stringent chemical regulations to less developed nations with more relaxed regulations. This shift may exacerbate the environmental harm caused by agricultural production in these less developed countries. The main environmental problems caused by agricultural trade liberalization are: first, increased greenhouse gas emissions and increased atmospheric pollution. Second, increased fertilizer and pesticide use and changes in land use. Third, deforestation expansion and reduced biodiversity. Fourth, increased water use and changes in water footprints.

Many studies have examined the negative impact of trade on the environment, with a particular focus on its first effects. Anselme et al. explored the correlation between trade and environmental quality in Sub-Saharan Africa (SSA) by utilizing the Environmental Kuznets Curve theory [45]. They found that trade had a significant impact on SSA's emissions of nitrous oxide, ACH4, and carbon dioxide, and concluded that trade consistently has a negative impact on the environment, regardless of wealth levels. The research conducted by Amogh et al. and Nosheen et al. supports the notion that trade liberalization has a direct and significant impact on carbon dioxide emissions [46,47]. In the short term, agricultural trade liberalization may have a positive impact on reducing agricultural environmental pollution by acting as a driving force. Zhang et al. conducted a survey of ASEAN countries and found that increased trade liberalization and energy use resulted in increased environmental damage [48]. Cemal investigated the interaction between trade and the environment in terms of carbon emissions in the trade liberalization of ASEAN countries and demonstrated an inverted s-shaped relationship between carbon dioxide emissions and the region [49]. Hakimi and Hamdi found that the long-term increase in carbon emissions in Tunisia and Morocco was linked to trade openness [50].

Numerous studies have focused on the environmental implications of free trade agreements and have concluded that the impact varies depending on the type of agreement. Specifically, when the agreement involves only developed or developing countries, there is no negative impact on the environment. In fact, such FTAs can even have a positive impact on the global environment. When agreements are made between developing and developed countries, this often leads to an increase in global greenhouse gas emissions, which can be harmful to the environment [51]. Saunders et al. conducted a study on bilateral trade between New Zealand and the European Union. The study found that trade liberalization resulted in an increase in greenhouse gas emissions on the New Zealand side, while the EU side experienced a decrease in emissions [52]. In their research, Yu et al. examined the impact of trade opening on GHG emissions in the United States and Mexico [53]. They discovered that after 1994, trade expansion resulted in a significant increase in GHG

emissions, with the majority of the change occurring in Mexico. Interestingly, the United States did not experience any effects from this trade expansion. Additionally, Lee and Zhang contend that trade liberalization has led to higher energy consumption and GHG emissions, particularly in developing countries with limited resources [54]. According to the authors mentioned earlier, Moon opposes the idea of free trade in agriculture [15]. He argues that free trade fails to account for the differences in countries and regions with regards to the environment.

In their research, Schmitz et al. examined the effects of trade liberalization on the environment using a global land use model [55]. They discovered that while further trade liberalization could lower global food costs, the majority of the economic benefits came at the cost of the environment and climate [55]. Similarly, Van et al. conducted studies that showed only minor changes in land use in Europe due to trade liberalization, but significant changes in Africa and other developing regions, resulting in detrimental environmental impacts [56]. Mittler and Knirsch argue that additional liberalization could have negative effects on the environment, particularly in developing countries, by exacerbating existing issues in key environmental sectors, such as forests and the marine environment [57]. Flachsbarth et al., in agreement with Lee and Zhang et al. [54], suggest that further trade liberalization may increase environmental pressures in certain regions of Latin America [58].

In their research, Huang et al. examined the impact of trade liberalization on rice production [59]. They discovered that the effect of trade liberalization on chemical intensity was significant on an annual basis [59]. Furthermore, they found evidence suggesting that chemical intensity could potentially increase in the context of free trade. In their paper, Longo and York disputed the notion that increased agricultural trade does not necessarily result in environmental damage, but rather can aid in reducing pollution [60]. The authors contended that greater exports of agricultural products lead to heightened use of fertilizers and pesticides worldwide. This finding calls into question policies that advocate for expanding agricultural trade liberalization. Trade liberalization in Asia may have also contributed to increased production within member countries, leading to heightened levels of domestic pollution due to the intensive use of agrochemicals [61,62]. According to Fei et al., agricultural trade liberalization has led to a significant increase in the domestic use of nitrogen, phosphate, potash, and compound fertilizers for the export of fruits, vegetables, and other crops (FFVs) from China [63]. The study found that the positive impact of this liberalization on domestic fertilizer use was more noticeable in the initial ASEAN countries than in those that had recently joined.

Schmitz et al. conducted a study in South Asia, Southeast Asia, and the Middle East, which revealed a correlation between increased trade liberalization and water scarcity, as well as a decrease in water prices [64]. On the other hand, Dang et al. argued that the impact of trade liberalization on total water use would only be modest [65]. They further noted that agricultural trade liberalization's effect on water use is dependent on the elasticity of water substitution. According to Konar, trade liberalization may result in a greater water footprint [66]. Additionally, in the French region, Neste trade liberalization was found to have negative impacts, such as decreased agricultural income and increased irrigation water usage [67].

Mark et al. discovered that trade liberalization could result in a rise in fish product prices and more advantages for fishermen due to an increase in the number of fishing boats [68]. Unfortunately, this could also lead to ecosystem degradation. Once subsistence income levels are attained, however, this may prompt fishermen to become more cautious in their use of resources or even reduce their fishing activities [68]. Rivera contends that export-oriented aquaculture and agriculture can drive economic growth under liberalization policies but may also cause harm to the local environment, including deforestation and mangrove degradation [69]. Supporting this argument, Schmitz et al. demonstrated that trade liberalization can lead to deforestation expansion in the Amazon, resulting in substantial carbon emissions [9]. According to Astier et al., the North American Free Trade Agreement (NAFTA) brought about significant changes in the agricultural system of rural

Mexico [70]. The authors argue that small farmers were unable to compete with global imports of maize products without price subsidies, which led to the selling or abandonment of their land. As a result, industrial agriculture expanded into previously untouched areas, such as forests, secondary vegetation, and staple crops.

Kirchner and Schmid employed scenario analysis to investigate the potential environmental impacts of agricultural trade policies in the Malchiefeld region of Austria [71]. Their findings revealed that a laissez-faire market scenario, which involved the removal of trade barriers and agri-environmental payments, resulted in significant environmental deterioration. Laura et al. found that agricultural trade liberalization had a negative impact on environmental utility in Switzerland, resulting in an 11% reduction [72]. This was primarily due to the effects on land use, which led to a decrease in biodiversity expression and an improvement in soil degradation, thanks to a better nutrient balance.

Furthermore, Rauf et al. discovered that trade liberalization is negatively impacting the environment in China [73]. This finding was supported by Chen et al., who developed a model using NASA air pollution data [74]. They argued that the expansion of trade has resulted in the deterioration of air quality in China, with the polluting sector and trade being the primary causes of environmental degradation. Li and Huang argue that China's growing international status and the liberalization of international trade have resulted in increased demand for agricultural products. However, this has led to severe damage to the agricultural environment, with the percentage of damage increasing from 21% in 2014 to 45% in 2018 [75]. While structural effects can help reduce fertilizer concentrations and their impact on the environment, when combined with other factors, such as agricultural trade, it can actually worsen agro-environmental pollution in China.

### 4.3. The Impact of Agricultural Trade Liberalization on Environmental Quality Is Unclear

Trade liberalization can have both positive and negative impacts on the environment of a region. According to Gumilang et al., Indonesia's involvement in AFTA and IJEPA agreements is unlikely to result in significant changes in environmental performance, and the impact is mixed at best [76]. This is because the environmental effects of trade liberalization are relatively minor due to the tariff reforms that are causing air pollution while also reducing water pollution. Coxhead and Jayasuriya contribute to environmental degradation by distorting agricultural incentives that encourage deforestation in the Philippine region [77]. However, it is important to note that the impact of trade liberalization on the environment can also be positive if the government takes appropriate actions. Verburg et al. conducted a study using multi-country data and models to examine the influence of agricultural trade on GHG emissions [78]. Their findings indicate that inter-regional transfers that occur during agricultural production and livestock breeding contribute to an increase in total global GHG emissions. However, the impact of these emissions varies across countries, with some experiencing an increase while others experience a decrease. According to Bourgeon, the impact of trade liberalization on global emissions can vary depending on the comparative advantage of the region [79]. It can either increase or decrease emissions compared to self-sufficiency.

### 4.4. The Environmental Impact of Agricultural Trade Liberalization Is Insignificant

Robert and his colleagues conducted a study to evaluate the potential effects of global agricultural trade liberalization on the agricultural industry in the United States [80]. Their findings suggest that the impact of agricultural trade liberalization on the environment is generally small, less than 1%, and falls within the grand scale of complete agricultural trade liberalization. Moreover, assuming that the environmental impact caused by trade shocks will fall within the average annual variation range, the impact on the environment is negligible. Kukla-Gryz, on the other hand, used a structural equation model and data from 120 developing countries to analyze the relationship between economic growth, trade, and the environment [81]. After considering the trade factor, Kukla-Gryz found that an increase in per capita GDP led to an increase in $CO_2$ emissions and fertilizer use density,

which in turn aggravated environmental pollution. Therefore, there is no evidence that trade promotes the improvement of environmental quality. Atici analyzes the relationship between $CO_2$ emissions per capita and three key factors in Central and Eastern European countries: GDP per capita, energy use per capita, and trade openness. By examining these variables, the author aims to shed light on the impact of economic development and trade on environmental sustainability in this region [82]. The study utilizes panel data from Bulgaria, Hungary, Romania, and Turkey spanning from 1980 to 2002. The findings support the existence of an EKC within the region, where per capita $CO_2$ emissions decrease as GDP per capita increases. Additionally, the results indicate that globalization, as measured by the trade openness variable, does not contribute to emission levels in the region.

### 4.5. Many Initiatives to Dispose of Negative Impacts

After reviewing the literature, it has become clear that agricultural trade liberalization could have negative impacts on the environment. The question now is how to address these impacts. The authors suggest that there are four key areas to focus on: factor distribution, policy adjustment, technological innovation, and compensation mechanisms.

#### 4.5.1. Factor Allocation

In order to mitigate the environmental consequences of agricultural trade liberalization, policymakers must prioritize the redistribution of agricultural factors and the enhancement of agricultural productivity. For instance, the Chinese government has taken steps to address the negative effects of excessive chemical fertilizer use in agricultural production, with the aim of achieving zero growth in such usage. This serves as an example of the kind of action that can be taken to reduce the environmental impact of agricultural practices [83].

#### 4.5.2. Policy Adjustments

The revisiting of trade policy approaches presents both challenges and opportunities. On the one hand, member countries can use trade-related measures to protect the environment, such as implementing national legislation to ensure that imports do not cause negative environmental effects, while adhering to WTO rules. On the other hand, to fully benefit from free trade, it is crucial to select appropriate environmental standards, integrate trade agreements with environmental policies, and prohibit imports that harm the environment or fail to meet environmental regulations [84]. Chakravorty et al. emphasized the importance of developing countries implementing inspection and enforcement mechanisms to mitigate the detrimental effects of globalization and free trade on the environment, specifically in relation to agricultural industrialization [85].

#### 4.5.3. Technological Innovation

Improved and effective technologies lead to increased resource efficiency and lower pollution effects [47]. In order to mitigate the negative environmental impacts of agricultural trade liberalization, it is essential for countries to adopt cleaner technologies, modern management measures, precise agricultural management, and sustainable development strategies [58]. This will improve the efficiency of resource allocation and contribute to a more sustainable future.

#### 4.5.4. Improving Compensation Mechanisms

In order to effectively address climate change mitigation and trade liberalization on a global scale, it is imperative to internalize environmental costs by factoring in the associated expenses of producing goods and taking into account environmental and climate externalities. Research conducted by Schmitz indicates that regions that experience a boost in trade can allocate a significant portion of the benefits towards compensating for any environmental damages incurred, such as deforestation and greenhouse gas emissions [55]. Brewer argues that trading countries must cooperate more to minimize instances where trade benefits countries but harms the environment [86]. As the majority of these costs are

borne by developing nations, compensation policies need to be established or enhanced. An example of an emerging compensation scheme is REDD+, which offers compensation to developing countries that commit to preserving their rainforests [87,88].

In addition to the aforementioned measures, it is highly recommended that governments prioritize increasing investments in climate and environmentally friendly technological advancements at the earliest opportunity. This will help alleviate the pressure on the land and the environment, ensuring a sustainable future for generations to come.

## 5. Conclusions

This study systematically reviews the literature of the past two decades and examines the effect of agricultural trade liberalization on the environment. While the relationship between the two is complex and debated by academics, the majority of the literature suggests that agricultural trade liberalization can impact environmental quality. Only three articles argue that the impact of agricultural trade liberalization on environmental quality is not significant.

The recent literature suggests that agricultural trade liberalization has a predominantly negative impact on the environment, resulting in reduced environmental quality due to increased greenhouse gas emissions, greater use of chemical fertilizers, excessive water consumption in agriculture, deforestation, and loss of biodiversity. However, some studies have also found that liberalization can have positive effects. For instance, it can enhance global resource allocation, improve efficiency in resource utilization, promote the use of more efficient technologies, and significantly decrease global environmental pollution.

The phenomenon of agricultural trade liberalization examined in this study is global, regional, and international in scope. However, its impact on the environment is typically felt in one specific country or region. Our research indicates that agricultural trade liberalization tends to have a more pronounced effect on the environmental quality of developing countries, particularly when it involves trade between developing and developed nations.

This study also delves into the mechanism behind the environmental impact of agricultural trade liberalization. It posits that the net environmental effect of such liberalization is the combination of four factors: the transport effect, scale effect, structural effect, and technology effect. However, in the absence of conditional constraints, the first three effects are generally negative, while only the technology effect is positive.

This study presents several potential solutions for reducing environmental pollution associated with trade liberalization by combining the findings and mechanisms of action. Firstly, there is a need to redistribute the factors of agricultural production. Secondly, policy adjustments need to be strengthened, and environmental standards and regulations should be implemented, along with the establishment of inspection and enforcement mechanisms. Thirdly, there is a need to boost technological innovation and its application. Finally, proposing to improve the compensation mechanism can be beneficial.

The impact of agricultural trade liberalization on the environment is multifaceted and influenced not only by a country's economic stage, but also by varying natural and agricultural conditions and comparative advantages. Consequently, this study proposes that to alleviate environmental pressures, governments worldwide must collaborate to coordinate trade agreements, enhance environmental standards, and establish an international trade framework that addresses ecological issues.

## 6. Research Outlook

This study provides a comprehensive overview of the environmental impacts of agricultural trade liberalization by drawing on the existing literature. While it has some limitations, it offers valuable insights for scholars and practitioners seeking to understand the relationship between trade liberalization and the environment. This study is sure to serve as a useful resource for future research in this field. In terms of trade and the environment, there are several research topics that could be explored in the future. The subjects include a wide range of environmental and economic topics, such as climate change, carbon

leakage, climate policy, environmental protection, air pollution, international environmental agreements, economic growth, carbon dioxide emissions, emissions trading, abatement costs, environmental performance, green supply chain management, composition effects, carbon footprints, and multi-regional input–output models [89]. To better understand the impact of agricultural trade liberalization on the environment, there are two areas of research that need to be explored.

Policy evaluation is a crucial tool for assessing the impact of agricultural trade liberalization on the environment. It involves evaluating the feasibility, effectiveness, and social cost of policies. One important method for policy evaluation is the use of system dynamic models (SDMs). SDMs allow for the simulation of the impact of different policies on both the environment and agricultural trade, providing a scientific basis for policy recommendations. By linking agricultural trade liberalization with various factors, such as the environment, economy, society, and policy, SDMs can evaluate the interaction and system effects among these factors. Furthermore, SDMs can assess the feasibility and impact of different strategies and provide corresponding policy recommendations.

A crucial issue that requires attention in the future is the selection of environmental standards. From an international standpoint, environmental externalities can be more effectively resolved through inter-country trade policies, specifically through negotiations on standards. Any unilateral approach to externalities will yield either insufficient or excessive regulation. In the former, consumers will overconsume products that cause environmental externalities. In the latter, although the externality can be improved, it will adversely impact exporting firms that adhere to this criterion. This underscores the importance of closely coordinating trade policies among countries, establishing unified norms and standards, and ensuring their implementation to achieve optimal outcomes when environmental external factors are present in trade.

**Author Contributions:** Conceptualization, P.W.; methodology, P.W.; software, P.W.; validation, P.W. and G.Q.; formal analysis, P.W.; investigation, P.W. and G.Q.; resources, P.W. and Z.R.; data curation, P.W. and Z.R.; writing—original draft preparation, P.W.; writing—review and editing, P.W. and G.Q.; visualization, P.W.; supervision, G.Q.; funding acquisition, G.Q. All authors have read and agreed to the published version of the manuscript.

**Funding:** This research was funded by the National Key R&D Program of China:"The Study on Collaboration of Supply Chain of Agriculture and Animal Husbandry between China and Mongolia" (2021YFE0190200) and the Graduate Research Innovation Project of Inner Mongolia:"Research on Supply Chain Collaboration of Agriculture and Animal Husbandry in China and Mongolia Based on Global Value Chain" (DC2000002099).

**Institutional Review Board Statement:** Not applicable.

**Informed Consent Statement:** Not applicable.

**Data Availability Statement:** Not applicable.

**Conflicts of Interest:** The authors declare no conflict of interest.

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
