# Peer review of "How Does Agricultural Trade Liberalization Have Environmental Impacts? Evidence from a Literature Review"

_sustainability, doi:10.3390/su15129379_

Round 1

Reviewer 1 Report

The manuscript is a systemic attempt to review the relevant literature on the impact of agricultural trade liberalisation. The strength of the manuscript is in its comprehensive analysis of two-decades of literature on the environmental impacts of agricultural trade liberalisation. The study proposes future researchers to consider about establishing a model to assess the environmental implications of trade liberalisaiton policies. I think the manuscript can be published in the current form. 

The quality of English language is good.

Author Response

Dear reviewer,Thank you very much for your hard work, taking the time and effort to review and consider our manuscript, and also for your recognition of our manuscript.Your comment has greatly benefited us. Thank you again.

Reviewer 2 Report

The aim of the paper is to study the impact of agricultural trade liberalization on the natural environment. The Authors referred to the functioning of the global food system and the challenges it has to face. These are, among others, the growing world population, the covid pandemic, frequent natural disasters and increased armed conflicts. The Authors drew attention to the important role of the liberalization of food flows in reaching deficit areas and improving the economic situation of many developing countries. At the same time, the strong connection between the growing food trade and its impact on the environment was underlined. The expansion of agricultural trade led to many positive and negative effects. The latter are the most crucial, such as: climate change, depletion and pollution of freshwater, eutrophication of river bodies, tropical deforestation, and loss of biodiversity. The Authors, through an extensive analysis of the literature, came to the conclusion that it is not obvious to define the mentioned impact unambiguously. There are many cases and much evidence indicating the positive influence of trade liberalization on the environment, but also negative one. Implementation of environmental agreements simultaneously at both national and international levels is crucial to achieving a positive impact. The Authors indicated many factors contributing to the improvement of the natural environment, including the promotion of sustainable production methods and the implementation of innovative solutions to improve production efficiency. At the same time, they pointed out that trade liberalization may deepen agri-environmental problems, especially in underdeveloped countries, where environmental standards are often of secondary importance and difficult to meet. Excessive use of natural resources, especially water for food production, is not without significance. The Authors suggest focusing on four key areas to deal with negative effects of liberalization of agriculture trade: factor distribution, policy adjustment, technological innovation, and compensation mechanisms. This study is valuable and due to its multifaceted and at the same time debatable character, is undoubtedly the basis for further research in the analyzed area. It should be pointed that the manuscript is presented in a well-structured manner. The cited references concerned mostly recent publications. The literature review is clear, comprehensive and relevant. A gap in the knowledge is appropriately identified and constitutes a fresh look at the analyzed area. I only noticed that there is no indication of the source under the figures.

Author Response

Dear reviewer,Thank you very much for your hard work, taking the time and effort to review and consider our manuscript, and also for your recognition of our manuscript.Your comment has greatly benefited us. Thank you again.

 Regarding the issue you mentioned "there is no indication of the source under the figures",we made the following response.

1.Figures of Row(48),We calculated based on the UN comtrade database.

2.Figures of Row(52),We calculated based on the UN comtrade database.

3.Figures of row(549),it comes from reference 74.

All the above have been modified in the paper.

Reviewer 3 Report

The manuscript ‘Does Agricultural Trade Liberalization Have Environmental Impacts?:Evidence From a Literature Review’ is interesting and useful. It is very important to know and understand the agricultural trade liberalization impacts (positive or negative) on the environment. I consider this brief review interesting for readers and merits publication.

Author Response

(The authors gave the same response as above.)

Reviewer 4 Report

The article is very well written and organized, easy to read and understand. In that sense, it is a very good job. The bibliography collected is extensive and interesting although not exhaustive for sure. Of course, the analysis circumscribes to specific areas and production systems, probably missing some relevant ones, such as open-sky grazing cattle production functions in the southern cone of South America.

The authors recognize that the topic is complex and multi-dimensional, and that "the connection between trade liberalization and environmental quality is highly intricate" (rows 51-52). As posed immediately, in rows 52-53: "Even today, there is no academic consensus on the impact of trade liberalization on the environment, and the debate continues". It should be recognized that the authors made an important effort to offer new contributions to the discussion.

Nevertheless, even with some disagreement, in the opinion of this reviewer, the recommendation is to accept the manuscript in its present form. The paragraphs that follow represent the opinion of this reviewer regarding the overall approach and, as an opinion, it was not considered to make this recommendation. They are made with the only intention to call the attention of the authors to some other ways to think about this problem. They do not pretend to make authors change their mind or their analytical approach either. Thus, this reviewer is not asking for a reply, he is just sharing some thoughts. The disagreement stated in what follows does not prevent recognizing the overall merits of the work.

The idea with which this reviewer ends the reading is that there is for sure a contribution, but the debate remains open. Nevertheless, in its conclusions, the article falls into a commonplace, frequently observed when these complex topics are addressed. That is, to suggest more and more public interventions, more regulations, more public policies, and so on. How does trade liberalization deal with more government influence on these issues? Is it really trade growth that impacts the environment, or it is government distortion? There is barely real free trade nowadays. What about the import quotas, high tariffs, non-tariff barriers, gov to gov trade, imposed mainly from developed to developing countries, which is the situation we really find today, mostly in agriculture?

It is the opinion of this reviewer that only looking at some alleged dichotomy between trade and environmental impact is not the happiest way to address these issues. It is a matter of trade-off. If we as humanity want to feed more than 8 billion people, many of them completely unable to fulfill their most basic needs, we should recognize, first, that it would be impossible without real free trade, with none or minimum government intervention. Governments tend to prioritize the interests of their own communities (ie., maintaining their costly and inefficient food production systems regardless the harm they may impose to foreign communities that produce better, more efficiently, and more environmentally friendly).

Any human activity has an impact. Only by recognizing that real collaboration and solidarity imply that free trade is essential for development, we may put an end to the debate, which does not mean cutting research allowing us to understand how to deal with natural resources and the environment. Then, we can then discuss how to do this with the highest efficiency and the lowest impact as possible on the environment.

Author Response

(The authors gave the same response as above.)

Reviewer 5 Report

The paper deals with an interesting topic, is well organized and written. However, I do not consider that the paper brings important contributions or novelties in terms of results or any policy proposition applied to agriculture and the environment.

Detailing the main points:

(1) Considering the quality of the journal Sustainability, indexed in WoS and with a high impact factor, my understanding is that approved papers must have a scientific relevant contribution in the current context.

(2) I did not identify any relevant contribution in terms of database, methodology, results or new applications for this area in this paper.

(3) The article is based on a well-developed systematic literature review, but does not have a relevant contribution to the area and does not lead the reader to new facts or further analysis.

Author Response

Dear reviewer,Thank you very much for your hard work, taking the time and effort to review and consider our manuscript, and also for your recognition of our manuscript.Your comment has greatly benefited us. Thank you again.

Based on your comments and suggestions, we have summarized the innovative points and included them in the paper. We believe that this paper not only examines the types of environmental impacts of agricultural trade liberalization, but also explores its impact mechanisms, providing a more comprehensive understanding of the relationship between various environmental effects. Additionally, solutions have been proposed from different perspectives to address the negative impacts of agricultural trade liberalization on the environment, making this study innovative and valuable.

Reviewer 6 Report

I reviewed the paper titled “Does Agricultural Trade Liberalization Have Environmental Impacts?” by Wang and Qiao. The study discovers an overview to address environmental impacts of agricultural trade liberalization using a systematic review of the relevant literature from the past two decades. However, Comments on the manuscript entitled are listed below.

1.    The paper title is not lined with the objective and conclusions. It is obviously that agriculture has environmental impact, whereas the title must be “How”. Please change the title to be in harmony with objectives and outcomes. Besides, since the paper is a review manuscript, it is suggested to mention that in the title.

2.    The introduction lacks references. For example, line 32-39 has no refence even though it is not the authors findings.

3.    Also, paragraph in line 40-49 has no refence. Please revise.

4.    There are some typos. E.g., do not use abbreviation in line 76 [it’s].

5.    Some used papers are not referenced. E.g., in line 347 [Karunakaran and Sadiq], in line 479 [Schmitz et al.], and in line 488 [Mark et al.].

Therefore, i recommend to return the paper to the authors for minor changes before proceeding to the publication.

Minor editing of English language required. Also, some of the used papers are not referenced. 

Author Response

Dear reviewer,Thank you very much for your hard work, taking the time and effort to review and consider our manuscript, and also for your recognition of our manuscript.Your comment has greatly benefited us.

Based on your comment, we have made the following modifications.
One is to change the title to "How Does Agricultural Trade Liberalization Have Environmental Impacts?: Evidence From a Literature Review".
The second is to include references in the introduction.
The third issue is the modification of abbreviations.
The fourth is to cite the literature that was forgotten to be cited in the previous text, and then cite it again.
Thank you again for your professional comments, which have made our article more scientific and normative.

Round 2

Reviewer 5 Report

I maintain the non-recommendation of this paper, detailing the main points:

(1) I did not identify any relevant contribution in terms of database, methodology, results or new applications for this area in this paper.

(2) The article is based on a well-developed systematic literature review, but does not have a relevant contribution to the area and does not lead the reader to new facts or further analysis.

Author Response

Dear reviewer, we would like to thank you for taking the time and effort to review our paper. Your professional and rigorous spirit is very worthy of our learning, and we also appreciate your questioning. After several days of discussion, all of our authors have replied to your two questions again.

1.In this paper, we summarized the relevant mechanisms of the environmental impact of agricultural trade liberalization and explored the relationship between various effects. At the same time, we drew a mechanism diagram, which we believe was lacking in the past.

2.Through literature review in this field, we believe that the selection of environmental standards and policy evaluation will be two hot research topics in the future